# The Effect of Hydrolyzed and Fermented Arabinoxylan-Oligo Saccharides (AXOS) Intake on the Middle-Term Gut Microbiome Modulation and Its Metabolic Answer

**DOI:** 10.3390/nu15030590

**Published:** 2023-01-23

**Authors:** Andrea Polo, Marta Acin Albiac, Alessio Da Ros, Vimac Nolla Ardèvol, Olga Nikoloudaki, Fabienne Verté, Raffaella Di Cagno, Marco Gobbetti

**Affiliations:** 1Faculty of Science and Technology, Libera Universitá di Bolzano, 39100 Bozen, Italy; 2Puratos NV, 1702 Dilbeek, Belgium

**Keywords:** arabinoxylan-oligosaccharides, SHIME, colon microbiome, short-chain fatty acids, lactic acid fermentation

## Abstract

Although fermentation and hydrolyzation are well-known processes to improve the bioavailability of nutrients and enable the fortification with dietary fibers, the effect of such pre-treatments on the prebiotic features of arabinoxylan-oligosaccharides (AXOS) had not been explored. The middle-term in vitro simulation through the Simulator of the Human Intestinal Microbial Ecosystem (SHIME) demonstrated that the feeding with different formulations (namely oat bran, rye bran and wheat bran) containing hydrolyzed AXOS fermented by lactic acid bacteria significantly increased the synthesis of short-chain fatty acids (SCFA) by colon microbiota, with hydrolyzed and fermented rye bran displaying the highest effect. After two weeks from the interruption of intake, SCFA concentrations significantly decreased but remained still significantly higher compared to the original condition. The microbiome was also affected, with a significant abundance increase in Lactobacillaceae taxon after feeding with all fermented and hydrolyzed formulates. Hydrolyzed and fermented rye bran showed the highest changes. The fungal community, even if it had a lower variety compared to bacteria, was also modulated after feeding with AXOS formulations, with an increase in *Candida* relative abundance and a decrease in *Issatchenkia*. On the contrary, the intake of non-hydrolyzed and non-fermented wheat bran did not produce relevant changes of relative abundances. After two weeks from intake interruption (wash out period) such changes were mitigated, and the gut microbiome modulated again to a final structure that was more like the original condition. This finding suggests that hydrolyzed AXOS fermented by lactic acid bacteria could have a more powerful prebiotic effect compared to non-hydrolyzed and non-fermented wheat bran, shaping the colon microbiome and its metabolic answer. However, the intake should be continuous to assure persistent effects. Opening a window into the ecological evolutions and plausible underlying mechanisms, the findings reinforce the perspective to explore more in depth the use of hydrolyzed and fermented AXOS as additional ingredient for bread fortification.

## 1. Introduction

The human intestinal ecosystem harbors a dynamic and complex community of microorganisms, both taxonomically and functionally. The interaction between microbiota and host significantly affects not only the host digestive tract, but it plays a key role also in many immunological and physiological responses with relevant effects in distinct body sites and systems. Many bacterial strains and the metabolites they release have been positively correlated to beneficial effects against several diseases such as diabetes, obesity, hyperoxaluria, ulcerative colitis, colon cancer, irritable bowel syndrome, neurodegenerative and cardiovascular diseases [1,2]. Consequently, the modulation of commensal gut bacteria is considered nowadays as a promising therapy to prevent or treat such problems [2]. In this scenario, the diet has a pivotal role driving the assembly and evolution of intestinal microbiota, determining both structural and functional properties, and it has been recently proposed as a potential strategy to shape the gut ecosystem in order to improve health status [1,2,3].

In last decade, natural prebiotics, mostly fibers, earned a considerable importance for human diet as they were linked with a plethora of beneficial effects leading to significant improvement in health status [4]. Even if the exact mechanism of action is not fully known yet, they are fermented by the gut bacteria changing the microbial composition and activity. The abundance of groups/species with capacity for using the variety of simple and complex possible structures of such carbohydrates as an energy source increase with resulting beneficial physiologic effect on the host [5]. The expression of such fermentative machinery in distinct species results in divergent specialization of gut microbiota [1]. Arabinoxylan oligosaccharides (AXOS) represent a class of dietary fibers with prebiotic properties deriving from arabinoxylans, complex carbohydrates found in the cell walls of the starch endosperm, the aleurone layer and in pericarp tissues of cereals [6]. The prebiotic properties and health effects (e.g., lowering cholesterol and reduction in the plasma ceramide levels) of AXOS have been widely described [4,6,7]. The production of short-chain fatty acids (SCFAs) due to AXOS fermentation is another determinant driving the improvement in the metabolic health [8,9]. Consequently, the inclusion of AXOS in the formulation of certain foods such as baked goods has been favored. For instance, endoxylanase enzyme has been used for more than 30 years to improve dough consistency, bread volume and crumb structure through the solubilization of arabinoxylans. By using the appropriate type and dose of enzyme, naturally present arabinoxylans fibers are converted into AXOS fragments, thus providing an in situ production approach that may be an alternative to straightforward fortification [10]. However, despite the increasing interest of the scientific community toward the promising role of AXOS, their potential has not been fully investigated. In particular, it is not clear how AXOS may affect intestinal ecosystem and metabolic health in the long-term period. They have the capability to shape the human gut microbiota [1] generating, for instance, a bifidogenic effect [4]. However, the literature about the overall effects of AXOS consumption did not show univocal data, such as their controversial effect on the abundance of lactobacilli [5,6,10]. Moreover, although fermentation and hydrolyzation are well-known processes to improve the bioavailability of AXOS and other nutrients, and to enable the food fortification with dietary fibers [11], the effect of such pre-treatments on the prebiotic features of AXOS from the most widely consumed main cereals, like oat, rye and wheat brans, was not previously explored.

Facing this non-univocal and incomplete background, we believe that in vitro experiments through gastrointestinal simulators represent a promising way to elucidate the effect of single food components as such an approach a priori excludes interferences from other factors like dietary habits and human physiology and avoids expensive and ethically complex human challenges [12]. In this study we aimed to investigate and compare the effect of three different formulations containing hydrolyzed AXOS from oat, rye and wheat brans fermented by lactic acid bacteria and a non-hydrolyzed/non-fermented AXOS formulation (as a control). To this purpose, the Simulator of the Human Intestinal Microbial Ecosystem (SHIME^®^) was adopted as it represents a scientifically validated and representative dynamic model of the human gastrointestinal tract in which the gut microbiota is reproduced from a fecal sample [13]. To the best of our knowledge this is the first study exploring the middle-term dynamics (over a 2 week timespan) deriving from the intake of hydrolyzed and fermented formulations containing AXOS obtained from three cereal flours, and it shows a window into the ecological evolutions and plausible underlying mechanisms.

## 2. Materials and Methods

### 2.1. AXOS Formulations

Four cereal-based flours containing AXOS were produced and supplied by Puratos company (Bijgaarden, Belgium): hydrolyzed oat bran fermented with lactic acid bacteria (MOB); hydrolyzed rye bran fermented with lactic acid bacteria (MRB); hydrolyzed wheat bran fermented with lactic acid bacteria (MWB); and non-hydrolyzed and non-fermented wheat bran (WB).

Preparation of MOB, MRB and MWB consisted of a mixture of water and 25% *w*/*v* of the corresponding bran and 10^7^ CFU/g of a proprietary *Lactiplantibacillus plantarum* strain. Mixture was fermented for 24 h at 30 °C. Initial pH values were 7.4, 6.4 and 6.4 for oat bran, rye bran and wheat bran, respectively, while final pH values were 6.6, 3.3 and 3.4 for MOB, MRB, and MWB, respectively. Subsequently, a commercial xylanase belonging to family 11 (GH11 xylanase, derived from *Thermopolyspora flexuosa*) was added at 7.5 g/kg bran and the mixture was incubated for 5 h at 70 °C in a ThermoMixer C (Eppendorf, Milan, Italy). Enzymatic activity was stopped by increasing temperature to 90 °C for 30 min. For the WB mixture only the enzymatic incubation and heat inactivation were performed. Heat inactivated formulations were lyophilized to obtain a dry powder. Total xylose (Xyl Tot), total xylose in water-extractable arabinoxylan (WEAX Xyl Tot), free xylose (Free Xyl), AXOS, glucose, insoluble, soluble and total dietary fibers, and arabinoxylan-xylose ratio (Ara/xyl) were quantified (Table 1) from the lyophilized samples according to (Courtin et al., 2000).

### 2.2. SHIME^®^ Four-Line Configuration and Experimental Design

The SHIME^®^ (ProDigest, Gent, Belgium) system configuration consisted of four identical units of bioreactors operating in parallel (4-SHIME^®^). Each 4-SHIME^®^ unit consisted of three double-jacketed vessels maintained at 37 °C under anaerobic conditions, simulating the stomach and small intestine (ST/SI), proximal colon (PC) and distal colon (DC), respectively [14]. PC and DC bioreactors contained 500 and 800 mL of adult L-SHIME^®^ growth medium (ProDigest, Gent, Belgium), respectively, simulating a digested feed. Ranges of pH values in PC and DC bioreactors were 5.6–5.9 and 6.6–6.9, respectively. PC and DC pH was computer controlled using 0.5 M HCl and 0.5 M NaOH to mimic the colon physiological conditions. The ST/SI reactors were set at pH 2. PC and DC vessels were inoculated using the fecal material of a healthy 38-year-old volunteer. Fecal slurry was prepared as reported by Molly et al. [15] and inoculated [10% (*v*/*v*)] within 1 h after collection. The 4-SHIME^®^ experimental design consisted of 14 days of stabilization period needed to shape and stabilize the microbiota composition of the donor’s fecal material in representative colon ecosystems. It was followed by 14 days basal period (control period) that represented the experiment baseline. During these periods, the ST/SI vessel had a fill-and-draw approach controlled by a temporal switch to first pump, 3 times/day (to simulate 3 daily meals), 140 mL of adult L-SHIME^®^ growth medium (ProDigest, Gent, Belgium) and 60 mL of pancreatic juice (12.5 g/L NaHCO_3_, 0.9 g/L pancreatin and 6 g/L oxgall) into the vessel and then transfer the mixture to the colon vessels. Feeding cycles happened at a fixed time schedule, thereby allowing an 8-h period of fermentation between each feeding cycle. During the control period, 3 times/week lumen samples were collected from PC and DC reactors 10 min before the start of next feeding cycle (which ensured there is no feed or colon content entering the adjacent colon vessels to dilute or interfere with metabolite quantification). SCFA (acetate, propionate and butyrate) in lumen samples were measured in order to control the fermentative activity of the microbial communities and assess the reproducibility and stability of gut ecosystems. Subsequently, a treatment period started in which the four products containing AXOS were supplemented to the basal feed medium in the respective SHIME^®^ units (one for each unit). The daily intake was 2.14 g for each assayed AXOS which has already been shown to exert beneficial physiological effects [16]. Finally, a 2-week wash out period without AXOS addition was carried out. During the 4-SHIME^®^ run, all bioreactors were continuously stirred, flushed once a day with sterile N_2_ and monitored for constant volume and pH stability.

During treatment and wash out, samples of lumen (40 mL) were collected at regular times, at the end of control period (T0), after one week of treatment period (T1) and at the end (i.e., after 2 weeks) of both treatment (T2) and wash out (T3) periods, from both PC and DC reactors (Figure 1). Lumen samples were stored at −80 °C until further analysis.

### 2.3. Fecal Donor Selection

For the inoculum of PC and DC vessels of the 4-SHIME^®^ model, a representative donor of fecal inoculum was selected from a cohort of typical Mediterranean diet (MD) consumers as described by [12]. To this purpose, an initial cohort of 61 healthy volunteers aged between 19 and 50 years, no evidence of pathologies, no history of drug use in the last 6 months, no smokers, no regular alcohol consumers, not pregnant, and who had been antibiotic and probiotic free in the 6 months prior to the sample collection was recruited with fully informed written consent from all volunteers. Nutritional questionnaires about dietary habits and frequencies of food consumption were administered [17]. Based on the questionnaires, the adherence of the whole cohort to the Mediterranean diet was assessed by calculating an MD score (MDS) which ranged from 0 to 8 points (minimal to maximal adherence) [3]. The cutoff of 4 was used to define a satisfactory adherence to MD [3]. Forty volunteers out of 61 showed an MDS from 4 to 8, and were recruited to collect fecal materials into sterile bags, added (ratio 1:2) of RNA later solution (Applied Biosystems, Foster City, CA, USA) at 4 °C. The mixture was immediately homogenized into sterile bags using a stomacher apparatus (Stomacher 400 Circulator, Seward, United Kingdom). Homogenized samples were stored at −80 °C until use. Such samples were analyzed for their microbiota composition and SCFAs content. The selection of the donor was based on the clustering of fecal microbiota abundances of the 40 recruited individuals, aggregated at family level, together with SCFA data. Clustering was performed with the Manhattan distance matrix and Ward D2 method [12]. Partial least-squares discriminant analysis (PLS-DA) was performed considering the adherence to MD as an independent variable and OUT abundances as feature for the model. The contribution of each feature was further explored and annotated with the explanatory independent variable level. All statistical analyses were performed in R programming version 4.04 (R Foundation, Vienna, Austria) [12]. The collection of data from consumers and the use of human fecal samples were approved by the Ethics Committee of the Free University of Bozen on 17 July 2019, and informed consent for the experimentation was obtained from all subjects involved in the study.

### 2.4. SCFA Quantification

Luminal samples from 4-SHIME^®^ vessels were centrifuged and supernatants were filtered with 0.2 µm filter (Whatman, Darmstadt, Germany). Extraction and analysis were performed as described by Lotti et al. [18]. A Trace GC Ultra gas chromatograph (Thermo Fisher Scientific, San Jose, CA, USA) coupled to a TSQ Quantum XLS tandem mass spectrometer (Thermo Fisher Scientific, San Jose, CA, USA) was used. The chromatographic separation was through a fused silica Stabilwax^®^-DA column (30 m × 0.25 mm i.d. × 0.25 μm) (Restek Corporation, Bellefonte, PA, USA). The MS detection operated on full-scan mode (EI at 70 eV, ion source temperature at 250 °C, m/z values ranged from 40 to 300 Da and acquisition scan time 0.2 s) and multiple reaction monitoring acquisition mode [18]. The GC-MS data processing was performed using the qualitative and quantitative software package XCALIBUR™ 2.2 (Thermo Fisher Scientific, San Jose, CA, USA). SCFA analyses were performed in triplicate, and the mean values, standard deviation and variance analysis with one-way ANOVA of all replicates were calculated using GraphPad Prism Differences were considered significant with *p*-values *p* < 0.05. Individual comparisons were made post hoc with the Tukey–Kramer test.

### 2.5. Microbial Community Analysis

Samples of lumen collected in 4-SHIME^®^ vessels were centrifuged (10,000 rpm for 10 min), and genomic DNA was extracted from the resulted pellet using the FastDNA Spin Kit For Soil (MP Biomedicals, Irvine, CA, USA). Each sample was extracted in duplicate. The DNA concentration was quantified by using a Nanodrop One/One Spectrophotometer (Thermo Fisher Scientific, San Jose, CA, USA) and amplified through polymerase chain reaction (PCR). The 16S rRNA variable region V3–V4 and internal transcribed spacer (ITS) region were used for bacteria and fungi, respectively. Amplicons were cleaned using the Agencourt AMPure kit (Beckman coulter, Brea, CA, USA), and DNA was quantified using the Quant-iT PicoGreen dsDNA kit (Invitrogen, Waltham, MA, USA). Quality and purity of the library was assessed with High Sensitivity DNA Kit (Agilent, Santa Clara, CA, USA) by the Bioanalyzer 2100 (Agilent, Santa Clara, US). Library preparation and pair-end sequencing were carried out at the Genomic Platform—Fondazione Edmund Mach (San Michele all’Adige, Trento, Italy) using the Illumina MiSeq system (Illumina, San Diego, CA, USA). Raw paired-end FASTQ files were demultiplexed using idemp (https://github.com/yhwu/idemp/blob/master/idemp.cpp, accessed on 7 June 2017). Sequences were quality filtered, trimmed, de-noised and merged using DADA2 pipeline [19]. In the case of yeast, PCR primers were removed using cutadapt (https://github.com/marcelm/cutadapt, released on 11 May 2022). Chimeric sequences were identified and removed using consensus method to derive Amplicon Sequence Variants (ASV). Taxonomy assignment was performed against SILVA database (v138.1) in the case of bacteria, while UNITE (v8.3) database was used for yeasts. Rarefaction curves were computed for all samples and then relative abundance was estimated.

### 2.6. Statistical Analyses

Beta diversity was assessed across time, 4-SHIME^®^ segment and feeding time through non-metric multidimensional scaling procedure, while alpha diversity within the sample was computed using different indices (Chao1, Gini Simpson, inverse Simpson, Shannon and Fisher). Prevalence of microbial genera on the aggregated core microbiome of proximal and distal 4-SHIME^®^ colon tracts was assessed at different sampling points and depending on the feeding type. Community structure variations when comparing different time points was determined by Rank Sum Wilcoxon test at different taxonomic levels (*p* < 0.05 and medial log change ranging from −10 to 10 fold change). Only significant variations were annotated on the tree structure.

Non-metric multidimensional scaling (NMDS) analysis was applied to highlight statistically significant differences for the bacterial microbiome composition, comparing before and after feeding and comparing among different AXOS formulations. Alpha diversity of microbiome in different colon tracts and at different time points for each thesis was determined with Chao1, Gini Simpson, inverse Simpson, Shannon and Fisher indexes.

Pearson correlation was established among initial content of Xyl Tot, WEAX Xyl Tot, Free Xyl, glucose, insoluble, soluble and total dietary fibers and AXOS in MOB, MRB, WB and MWB, and microbial abundance and SCFA concentration in luminal samples.

## 3. Results

### 3.1. Fecal Donor Representative of High Adherence to Mediterranean Diet

Based on the clustering analysis, the fecal donor was randomly selected as a representative of the main cluster of volunteers displaying high adherence to MD (MDS from 4 to 8).

### 3.2. SCFA Profiles

Before feeding with treatment (T0), no significant (*p* > 0.05) differences were found among 4-SHIME^®^ units in both colon tracts, for all investigated SCFA (Table 2). However, the consumption of different AXOS formulations enhanced the synthesis of SCFA at the colon level and resulted in significant (*p* < 0.05) differences in SCFA concentrations both among different formulations and during time (T1 and T2). A significant (*p* < 0.05) increase was observed already after one week of treatment (T1), both in PC and DC reactors. The intake of MRB resulted in the highest concentration of acetic and butyric acids, significantly (*p* < 0.05) higher than those measured after feeding with MOB, WB, and MWB (Table 2). The highest (*p* < 0.05) concentration of propionic acid was found after feeding with WB. In general, after two weeks of wash out period (T3) a significant (*p* < 0.05) decrease in SCFA concentration was found compared to the treatment period for all SCFA and in all 4-SHIME^®^ units (fed with different formulations). The only exceptions were found for acetic acid in PC fed with MOB, which was significantly (*p* < 0.05) higher compared to previous time points and compared to other 4-SHIME^®^ units at the same time point, butyric acid in PC fed with MOB, which did not differ significantly (*p* > 0.05) compared to values measured during treatment period and was significantly (*p* < 0.05) higher compared to other 4-SHIME^®^ units at the same time point, and propionic acid in PC fed with MWB that showed no significant (*p* > 0.05) differences compared to values obtained at the end of treatment period. Overall, in both PC and DC reactors values after two weeks of wash out period (T3) were still significantly (*p* < 0.05) higher than those measured before treatment (T0), the exception being the concentration of acetic acid in PC fed with WB that had no significant (*p* > 0.05) difference. In PC tract, the highest concentration after wash out period were found in the 4-SHIME^®^ unit fed with MOB (53.1 ± 2.7, 4.0 ± 0.1 and 14.2 ± 0.2 mM for acetic, butyric and propionic acid, respectively), while in DC tract the highest concentration at T3 was observed in the 4-SHIME^®^ unit fed with MWB for acetic and butyric acids (52.3 ± 2.5 and 4.4 ± 0.1 mM, respectively), and in the units fed with MOB and WB for propionic acid (14.8 ± 0.5 and 15.8 ± 0.2 mM, respectively).

### 3.3. Colon Microbiome

The relative abundance at genus level of the yeast microbiome determined before, during and after treatment with the four formulations in the proximal and distal colon tracts, revealed *Candida* and *Issatchenkia* as the only two identified genera (Figure 2). Before treatment, *Issatchenkia* was dominant both in PC (ca. 98%) and DC (ca. 76%) followed by *Candida* (ca. 2 and 24% in PC and DC, respectively). The intake of MOB and MWB greatly affected the structure of fungal community enhancing the relative abundance of *Candida*, which became dominant both in PC (reaching ca. 99%) and DC (reaching ca. 95%). MRB intake also enhanced the relative abundance of *Candida* compared to T0 (reaching percentages between ca. 60 and 63% in PC and DC, respectively, at T1), even if the effect was less pronounced. However, in these three 4-SHIME^®^ units *Issatchenkia* became again dominant after two weeks of wash out period, and the relative abundance of *Candida* decreased almost to the original condition (between ca. 10 and 25% in PC and between ca. 1 and 4% in DC). On the contrary, WB intake did not affect the original community.

The relative abundance at family and genus level of the bacterial microbiome, before, during and after treatment with MOB, MRB, MWB and WB in the proximal and distal colon tracts was provided (Figure 3). Before treatment, 15 bacterial families were identified. Bacteroidaceae, Enterobacteriaceae, Lachnospiraceae and Veillonellaceae were the most abundant taxa in the aggregate microbiome of PC, while Bacteroidaceae and Lachnospiraceae were the most abundant in DC reactors. An increase in relative abundance of Lactobacillaceae and Veillonellaceae taxa and a decrease in Bacteroidaceae family were found in proximal colon tract after feeding with MOB, MRB and MWB. In addition, the relative abundance of Enterobacteriaceae decreased only after feeding with MRB and MWB, while the relative abundance of Lachnospiraceae decreased only after feeding with MRB. On the contrary, the feeding with WB only caused an increase in relative abundance for Lachnospiraceae family. In distal colon, an increase in Lactobacillaceae relative abundance and a decrease in Akkermansiaceae relative abundance were observed only after feeding with MRB. No evident changes were observed after feeding with MOB, MWB and WB. After the wash out period the bacterial communities came back to around the original structure both in PC and DC, with only Lactobacillaceae family remaining still higher only in the 4-SHIME^®^ unit fed with MRB. As the variety of bacterial community was considerably higher than that of fungi, further analyses were focused only on bacterial communities.

NMDS analysis was applied to highlight statistically significant differences for the bacterial microbiome composition comparing before and after feeding and comparing among different AXOS formulations (Figure 4a). Alpha diversity within the samples determined through Chao1, Gini Simpson, inverse Simpson, Shannon and Fisher indexes of microbiome in different colon tracts and at different time points for each formulation was reported in Figure 4b. Feeding with all four formulations significantly modulated the bacterial communities in PC tract compared to T0, and such changes were already evident after one week of treatment. Communities in PC reactors fed with MWB and MRB grouped in the same cluster while those fed with MOB and WB grouped in separate clusters. After the wash out period, communities that were fed with MRB and MWB came back to around the original condition, while the changes were more persistent for those fed with MOB and WB. On the contrary, all bacterial communities (both before, during and after treatment with different formulations containing AXOS) in DC tracts grouped in the same cluster, the exception being those fed with MRB that grouped in a separate cluster (Figure 4a). After the wash out period, also these communities were again like those before treatment.

To investigate the bacterial changes in more depth, the aggregate core microbiome at genus level was assessed at different detection thresholds, before, during and after feeding with different formulations (Figure 5). Before feeding, *Bacteroides*, *Lachnoclostridium*, *Klebsiella* and *Veillonella* were the prevalent taxa, followed by *Akkermansia*, *Blautia*, *Subdoligranulum* and *Bilophila*. Although *Lactiplantibacillus*, *Levilactobacillus, Lacticaseibacillus* and *Lentilactobacillus* genera were part of the core microbiome their relative abundances were very low with respect to the previous taxa. The relative abundances of *Bifidobacterium* and genera belonging to lactic acid bacteria significantly increased after feeding with all hydrolyzed and fermented formulations. In particular, after feeding with MOB *Lactiplantibacillus* and *Lentilactobacillus* taxa become significantly more abundant compared to T0, although the six most prevalent taxa did not change. After feeding with MRB, *Levilactobacillus, Lactiplantibacillus*, and *Lentilactobacillus* entered in the six most prevalent taxa, followed by *Bifidobacterium* genus that also increased significantly. On the contrary, *Lachnoclostridium*, *Klebsiella*, *Bilophila* and, especially, *Akkermansia* significantly decreased their relative abundance. After feeding with MWB, *Bifidobacterium* and *Agathobacter* became part of the six most abundant taxa. *Lactiplantibacillus*, *Lacticaseibacillus* and *Lentilactobacillus* also significantly increased their relative abundance while a decrease in *Klebsiella* and *Akkermansia* was observed. Differently, the feeding with WB did not increase the relative abundance of lactic acid bacteria. In this case, *Agathobacter* became part of the six most prevalent taxa. *Dialister* and *Fusobacterium* also showed a significant increase in relative abundance, while *Veionella* significantly decreased becoming one of the lowest abundant groups. After the wash out period (T3) bacterial communities modulated again becoming more similar to the original condition (T0), but some changes still persisted after two weeks. Mainly, in 4-SHIME^®^ line fed with MRB *Levilactobacillus, Lactiplantibacillus* and *Lentilactobacillus* were still significantly more abundant compared to T0, and in the 4-SHIME^®^ line fed with WB *Dialister* genus persisted even after two weeks of treatment interruption.

The taxonomic structure variations in bacterial community at different levels were shown in Figure 6, Figure 7, Figure 8 and Figure 9, comparing before and after feeding with each different formulation (panels A) and before feeding against the end of wash out period (panels B). Node sizes in these figures show the rarefied ASV counts, and colored nodes indicate a significant (*p* < 0.05) fold change for a given taxon. It demonstrated that Aeromonadaceae, Desulfobacterota (namely *Biophila*), *Centipeda* and *Sutterella* taxa were significantly (*p* < 0.05) more abundant before than after feeding with MOB. On the other hand, the treatment with MOB supported a significant (*p* < 0.05) increase in Lactobacillaceae (in particular, those belonging to *Lactiplantibacillus* and *Lentilactobacillus* genera) and Alphaproteobacteria (Figure 6A). Even after the wash out period, Lactobacillaceae (in particular, those belonging to *Lentilactobacillus* genus) remained significantly (*p* < 0.05) more abundant compared to the original condition, while *Veionella parvula*, Aeromonadaceae, *Sutterella* and *Lachnoclostridium* were still significantly (*p* < 0.05) more abundant before treatment (Figure 6B). A significant (*p* < 0.05) increase in Lactobacillaceae (in particular, those belonging to *Lactiplantibacillus*, *Lacticaseibacillus* and *Levilactobacillus* genera) was observed also after the feeding with MRB compared to before (Figure 7A). The treatment with MRB also caused a significant (*p* < 0.05) increase in *Veillonella atypica* abundance, while *Veillonella parvula*, Enterococcaceae, Enterobacteriaceae (*Klebsiella* genus), *Achromobacter*, Desulphobacteriota (*Bilophila* genus), Alphaproteobacteria and *Bacteroides xylanisolvens* taxa significantly (*p* < 0.05) decreased their abundance after the treatment compared to T0. After the wash out period, *Lacticaseibacillus* persisted with a significantly (*p* < 0.05) higher abundance compared to T0 (Figure 7B). Bacteroidaceae also were present with higher abundance compared to the original condition; on the contrary, *Bifidobacterium longum*, Aeromonadaceae, Burkholderiales (*Achromobacter*), Alphaproteobacteria (*Brucella* and *Ochrobactrum* genera) and Enterococcaceae had a significantly (*p* < 0.05) higher abundance before feeding (Figure 7B). The feeding with MWB also caused a significant (*p* < 0.05) increase in Lactobacillaceae (*Lactiplantibacillus*, *Lacticaseibacillus* and *Lentilactobacillus* genera) compared to before (Figure 8A). *Agathobacter* genus also increased significantly (*p* < 0.05). On the contrary, Enterococcaceae, *Klebsiella oxytoca*, Alphaproteobacteria (namely *Brucella* and *Ochrobactrum* genera), Desulphobacteriota (*Bilophila* genus) and *Lachnoclostridium* taxa were present with significantly (*p* < 0.05) higher abundances before the treatment. However, Lactobacillaceae did not persist with increased abundance after the wash out period. In this case, only *K. oxytoca* was still present with significantly (*p* < 0.05) higher abundance compared to T0, while Firmicutes and *Lachnoclostridium* were the only taxa showing significantly higher abundance at T0 (Figure 8B). The feeding with WB displayed a different scenario. The treatment supported a significant (*p* < 0.05) abundance increase in *Megaspaera*, Lachnospirales (namely *Agathobacter* and *Lachnoclostridium*) and *Bacteroides* taxa, while *V. parvula*, Lactobacillales (namely *Enterococcus faecalis*), *Aeromonas caviae*, *Brucella* and *Ochrobactrum* taxa were significantly (*p* < 0.05) more abundant before treatment (Figure 9A). After the wash out period *Megaspaera* and *Bacteroides* genera were still significantly (*p* < 0.05) more abundant compared to the original condition, while *V. parvula*, Bacilli, *A. caviae*, *Stenotrophomonas maltophilia* and *Brucella* were significantly (*p* < 0.05) higher at T0 (Figure 9B).

All sequencing results are publicly available at EML/NCBI/DDBJ under the accession number PRJNA868175 (https://www.ncbi.nlm.nih.gov/bioproject/868175, registration date 10 August 2022).

### 3.4. Synergy among AXOS Composition, Microbiome, SCFA and Dietary Fibers

Putative negative/positive synergy links among microbiome, SCFA and fibers in luminal samples, together with the initial composition of AXOS samples, were established through Pearson correlation between these two datasets. Statistically significant (*p* < 0.05) correlations among initial content of total xylose, total xylose in water-extractable arabinoxylan, free xylose, AXOS, glucose, SCFA, insoluble, soluble and total dietary fibers, and microbial abundance in luminal samples after 2 weeks of treatment period are shown in Figure 10a. SCFA, sugars and AXOS were significantly (*p* < 0.05) correlated with microbial groups (Figure 10b). The analysis demonstrated that lactobacilli positively correlate with acetic acid, which increased after MOB, MRB, MWB and WB fermentation by gut microbiota (Table 2), and glucose concentration in lumen. *Bifidobacterium*, *Aeromonas* and *Veillonella* genera negatively correlated with propionic acid, while *Klebsiella* negatively correlated with butyric acid. *Klebsiella* and *Brucella* genera positively correlated with xylose and AXOS, whereas *Enterococcus* and *Collinsella* with total xylose and acetic acid, respectively. No significant correlations emerged between the residual content of dietary fibers in lumen (soluble, insoluble and total) and residing microbial taxa. Considering the correlation between fibers and SCFA, only the soluble fiber had a significant (*p* < 0.05) and negative correlation with propionic acid.

## 4. Discussion

Based on the clustering analysis, we randomly selected a fecal donor from the main cluster of volunteers displaying high adherence to MD (MDS from 4 to 8). This allowed the selection of a highly representative fecal donor and, consequently, the evolution within the 4-SHIME model of representative gut ecosystems. When the 4-SHIME model reached the steady state, we investigated the ecological evolution of microbial communities in colon ecosystems after feeding the SHIME lines with three different formulations containing hydrolyzed AXOS from oat, rye and wheat brans fermented by lactic acid bacteria, and a non-hydrolyzed/non-fermented AXOS from wheat bran. The effects on microbiota inhabiting different colon tracts were fully characterized. Overall, the variety of bacteria resulted considerably higher compared to that of fungi, which included only *Candida* and *Issatchenkia* genus representing well-known groups in human gut [20,21]. Consequently, the attention was focused mainly on bacterial communities. Overall, they showed some common tracts: for instance, the relative abundance of *Bifidobacterium* increased after the intake of all formulations containing AXOS, as it is known to be a highly efficient AXOS utilizer [4]. However, in proximal colon tract, bacterial communities were also modulated differently by the intake of different formulations containing AXOS. The feeding with fermented and hydrolyzed formulations resulted in an increase in Lactobacillaceae and Veillonellaceae abundance and a decrease in Bacteroidaceae families, with MRB showing the highest changes. Differently, if fed with non-hydrolyzed/non-fermented AXOS, the bacterial community modulated following a different behavior. In this case, the relative abundance of Bacteroidaceae (namely *Bacteroides* species) was not subjected to reduction thanks to extensive glycolytic genomes enriched in xylanolytic genes that enable them to breakdown wheat bran [22], and an increase in *Lachnoclostridium* was observed [23]. Conversely, Bacteroidaceae family showed a decrease after feeding with all fermented and hydrolyzed formulations, probably due to the degradation of wheat, oat and rye bran during previous fermentation and hydrolyzation processes, while *Veillonella* species, which require lactate produced by other microorganisms for growth but are unable to metabolize normal dietary carbohydrates, increased their relative abundance as a result of a dynamic microbial succession that follows the significant increase in Lactobacillaceae abundance [24]. Similarly, the fungal community also evolved significantly after supplementation of fermented and hydrolyzed AXOS with an increase in *Candida* relative abundance and a decrease in *Issatchenkia*, while the intake of non-hydrolyzed/non-fermented AXOS did not produce relevant changes. This observed reshaping of both bacterial and fungal communities within the PC is not surprising since a large body of literature exists showing how bacteria interplay fungal traits in gut ecosystems and vice versa [25]. In any case, this study highlights how the preliminary fermentation by lactic acid bacteria and the hydrolyzation of cereal-based flours containing AXOS had the capacity to address the microbiome modulation within the PC tract in a different way compared to that given by non-fermented/non-hydrolyzed formulation. The enhanced relative abundance of Lactobacillaceae resulting after the intake of fermented and hydrolyzed formulations can be considered a beneficial effect. The mechanisms by which several species belonging to Lactobacillaceae family prevent the growth of pathogenic bacteria or fungi through nutrient or niche competition and/or the production of antimicrobial compounds has been largely demonstrated and aids the restoration of the gut microbiota after perturbation with antibiotics [26]. On the other hand, several yeasts belonging to *Candida*, which increased the relative abundance after feeding with fermented and hydrolyzed formulations, are considered true symbionts of the human gut. Certainly, it must be considered that the most frequently detected *Candida* species in feces of healthy humans is *Candida albicans*, a potential pathobiont of the human intestine [20]. Unfortunately, the analysis did not reach the species level, avoiding an accurate conclusion for the yeast. In this case, further research and analysis are desirable to provide more insights about the increase in this genera.

The NMDS demonstrated that changes induced in gut microbiome by the feeding with cereal-based flours containing AXOS are mitigated in distal colon tracts as almost all bacterial communities (both before, during and after intake of different formulations) grouped in the same cluster. Both reversible and more stable effects on the community depending on the different colon tracts were already reported [27].

In general, during the wash out period the gut microbiome modulated again to a final composition (after 2 weeks from intake interruption) that is more like the original condition, even if some variations still persist.

Given this scenario, we have explored the possible underlying biological mechanisms that determinate the observed evolutions by reviewing the available literature. The effects on gut microbiome referred by this study and by previous literature are non-univocal [5,6,10]. Considering that the most important drivers for the ecological evolution of a microbial population are environmental (chemical and physical) conditions and the metabolic capacity of involved species to use substrates for growing [22], the non-univocal results can be explained because the outcomes from the feeding with cereal-based flours containing AXOS are a result of complex competitions among different microorganisms with overlapping abilities to utilize such fibers. Moreover, it is known that ecosystems harboring distinct microbial communities reveal divergent responses to the same environmental input, adding further complexity [1]. Consequently, the comparison between our results and those available in literature confirms that the modulation of gut microbiome following the intake of AXOS can vary depending on the involved compounds and on the original gut ecosystem.

Considering the relativity of structural changes, we integrated metagenomics data with metabolic investigations to explore the effect of different types of AXOS-containing formulations on the microbiota activity. The analysis focused on acetate, propionate and butyrate as they are the main SCFA metabolites deriving from dietary fibers, and they are known to play a key role mediating the metabolic effects on hosts [5]. The SCFA profiles reflect the changes of nutritional conditions by colon microbiota and an expansion of the microbial population. In fact, the in vitro simulation through the 4-SHIME^®^ model demonstrated that overall, the feeding with all different formulations generated a significant increase in SCFA concentrations, with hydrolyzed and fermented rye bran displaying the highest effect. As the daily dose of AXOS was the same for the different formulations (2.14 g/day), this last finding may be due to the specific composition of MRB [28]. However, the new feeding regime shaped the microbial communities favoring species with metabolic capacity to break down the substrates containing AXOS and efficiently assimilate the released monomers, converting energy into new biomass. This ecological dynamic resulted in the increase in relative abundance by favored species (e.g., those belonging to *Lacticaseibacillus* genus after feeding with all fermented and hydrolyzed formulations), and on the consequent enhancement of SCFA concentrations [22]. After two weeks from the interruption of formulations intake, SCFA concentrations significantly decreased compared to those found at the end of the treatment period, but they still remained significantly higher compared to the original condition, confirming that the modulations of the microbial metabolic activities still persist after two weeks from intake interruption. Such a finding agrees with previous studies through both SHIME^®^ system [27] and randomized, double-blind, controlled, cross-over in vivo study [29]. Acetic acid by colonic fermentation, which showed the highest concentrations (specially upon MRB feeding), is known to cross the blood–brain barrier and act as an appetite suppressant in hypothalamus, thus suggesting a potentially relevant role in the management of metabolic diseases including obesity and diabetes [30].

A combined evaluation through Pearson correlation of gut microbiome, its metabolic answer and the initial composition of ingested AXOS formulations, allowed the exploration of the potential synergy links between the formulations composition and the effects on the gut ecosystem. A positive correlation between lactobacilli and acetic acid, which increased after MOB, MRB, MWB, and WB fermentation by gut microbiota, and glucose concentration in lumen was found. The links between microbiota diversity, SCFAs synthesis by colonic fermentation and glucose homeostasis were recently reviewed [31]. In this context, several studies showed the health-promoting features of fibers on patients mainly in terms of increased insulin sensitivity, reduced fasting, and postprandial glucose. SCFAs, as acetate, exert effects both at the local intestinal level and at a systemic level, acting through epigenetic mechanisms and via interaction with several receptors and tissues involved in the maintenance of glucose homeostasis.

Butyric acid is one of the most important metabolites in the gut microbiota [32]. Pearson correlation highlighted its negative correlation with *Klebsiella*, a genus routinely found in the human gastrointestinal tract as normal residing microbiota but including also some species behaving as opportunistic human pathogens. This finding agrees with previous studies in which the decrease in butyric acid in human gut was associated with the development of inflammations [33]. The butyrate metabolite can have the capacity to restore the intestinal barrier function under inflammatory conditions [34], being relevant in inflammatory bowel diseases where intestinal epithelial healing is an important therapeutic target. The positive correlation between *Klebsiella* and *Brucella* genera and xylose may be due to their capability to metabolize the compound [35]. Considering the positive correlation that they displayed also with AXOS, it can be hypothesized that these compounds have a prebiotic effect on these genera.

To the best of our knowledge, no previous evidence is available in literature about both the negative correlation between *Bifidobacterium*, *Aeromonas* and *Veillonella* genera with propionic acid and the positive correlation between *Enterococcus* and *Collinsella* taxa with total xylose and acetic acid that were found in this study. We hypothesize that these correlations may reflect the result of the specific competition among microbes which reach in different ways to the new environmental conditions that established after the intake of AXOS-containing formulations [1].

Overall, the findings of this study indicate that sustained consumption of different formulations containing AXOS, especially those derived by rye bran, orchestrates a wide variety of changes in proximal colon ecosystem, affecting both the microbiome and the metabolic activity in agreement with previous studies based on metagenomics, lipidomics, and metabolomics data integration [4]. Although the presented in vitro experiment and the hypotheses are a simplification compared to the continuous and variable dietary influx and changing host conditions to which gut bacteria are subjected in vivo, to the best of our knowledge this is the first study exploring the middle-term dynamics deriving from the intake of hydrolyzed and fermented formulations containing AXOS obtained from three cereal flours, and it provides a window into the ecological evolutions and plausible underlying mechanisms. Our findings open the perspective to explore more in depth the use of hydrolyzed and fermented AXOS as an additional ingredient for bread fortification, while transcriptomics studies will be necessary to depict the mechanisms of action.

## Figures and Tables

**Figure 1 nutrients-15-00590-f001:**
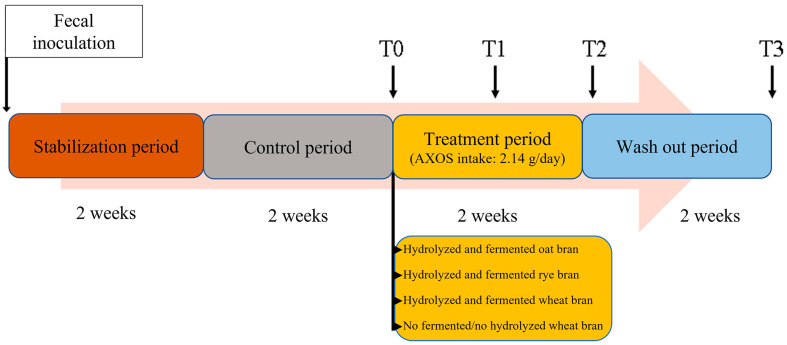
4-SHIME^®^ experimental design. T0–T3 represent sampling points. T0 is just before the start of treatment; T1 and T2 correspond to 7 and 14 days of treatment, respectively; and T3 corresponds to 14 days of wash out.

**Figure 2 nutrients-15-00590-f002:**
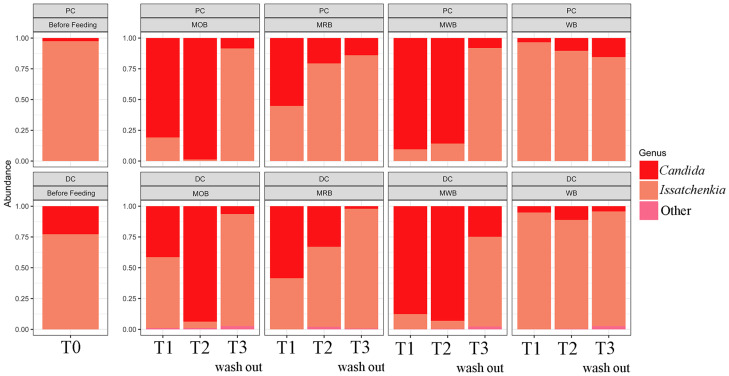
Relative abundance at genus level of the yeast microbiota, before (T0), during one (T1) and two (T2) weeks of feeding with MOB, MRB, MWB and WB, and after two weeks of wash out period (T3), in the PC and DC tracts.

**Figure 3 nutrients-15-00590-f003:**
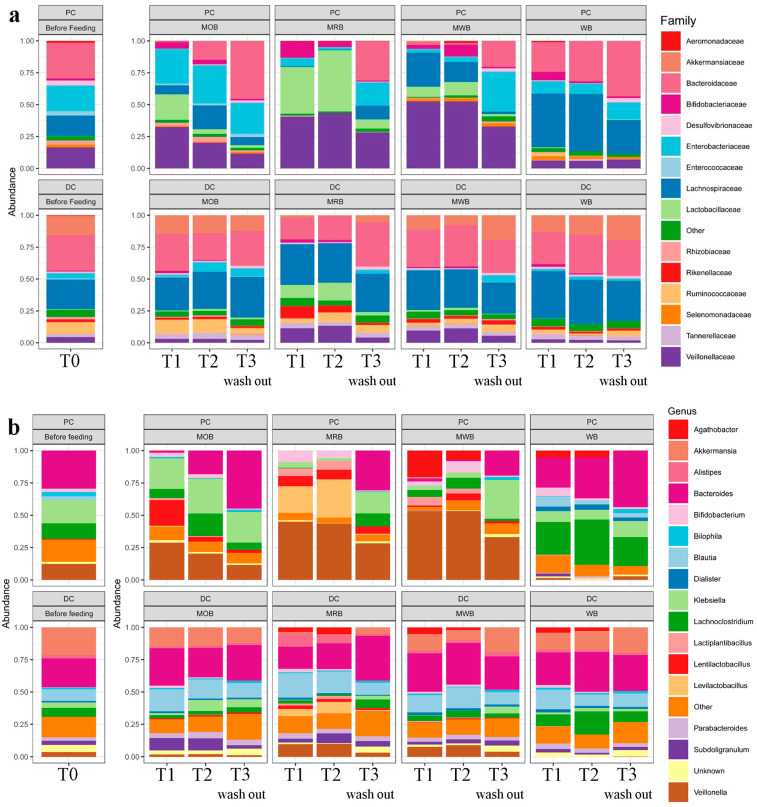
Relative abundance at family (panel **a**) and genus (panel **b**) level of the bacterial microbiota, before (T0), during one (T1) and two (T2) weeks of feeding with MOB, MRB, MWB and WB, and after two weeks of wash out period (T3), in the PC and DC tracts.

**Figure 4 nutrients-15-00590-f004:**
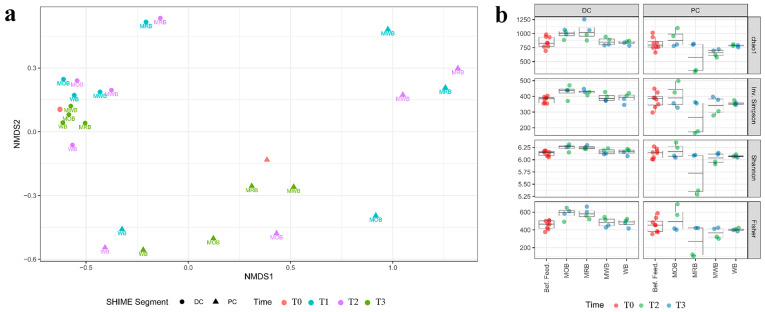
NMDS analysis (**a**) of the beta diversity within sample microbiota based on Bray–Curtis distance matrix of the luminal 4-SHIME^®^ samples before feeding (T0), during 1 (T1) and 2 (T2) weeks of feeding with MOB, MRB, MWB and WB, and after the wash out period (T3). The analysis considered sample inter-diversity of both PC and DC tracts. Alpha diversity (**b**) within the samples determined with Chao1, Gini Simpson, inverse Simpson, Shannon and Fisher indexes of microbiota in different colon tracts and at different time points for each formulation.

**Figure 5 nutrients-15-00590-f005:**
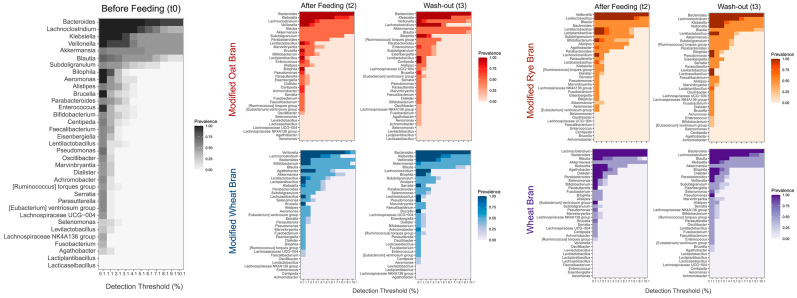
Pseudo-heatmaps showing the prevalence of aggregated core bacterial microbiota at genus level before (t0) and after two weeks (t2) of feeding with different formulations, and after two weeks of wash out period (t3). The color scales (gray for t0, red, orange, purple and blue for SHIME^®^ units fed with MOB, MRB, MWB, and WB, respectively) indicate the prevalence of OTU abundances across different detection thresholds ranging from 0.1 to 10.1%. For each pseudo-heatmap the term “modified” before the kind of AXOS means hydrolyzed and fermented by lactic acid bacteria.

**Figure 6 nutrients-15-00590-f006:**
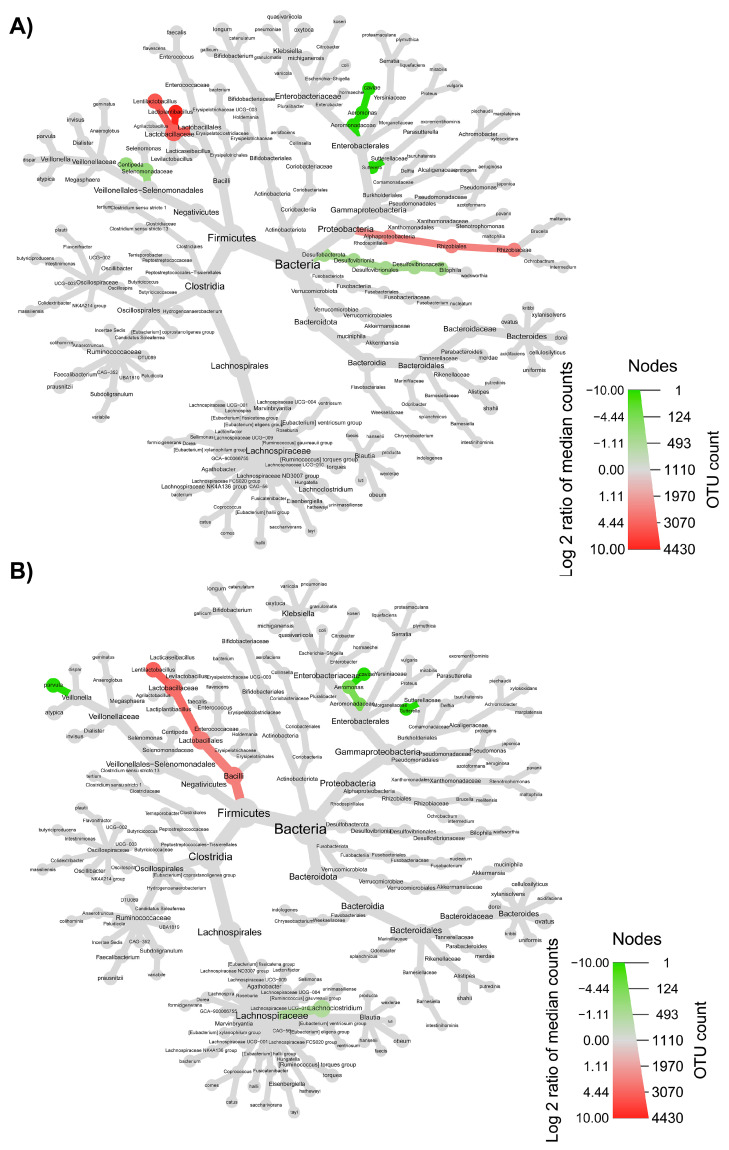
Microbial community taxonomic structure variations at different levels, before compared to after feeding with MOB (**A**), and before feeding against the end of wash out period (**B**). Node size denotes the rarefied ASV count, and colored nodes indicate a significant (*p* < 0.05) fold change for a given taxon: green color means that the taxon is differentially abundant before the treatment, while red color indicates that the taxon is differentially abundant after feeding (**A**) or after wash out period (**B**). Color intensity is proportional to the fold change.

**Figure 7 nutrients-15-00590-f007:**
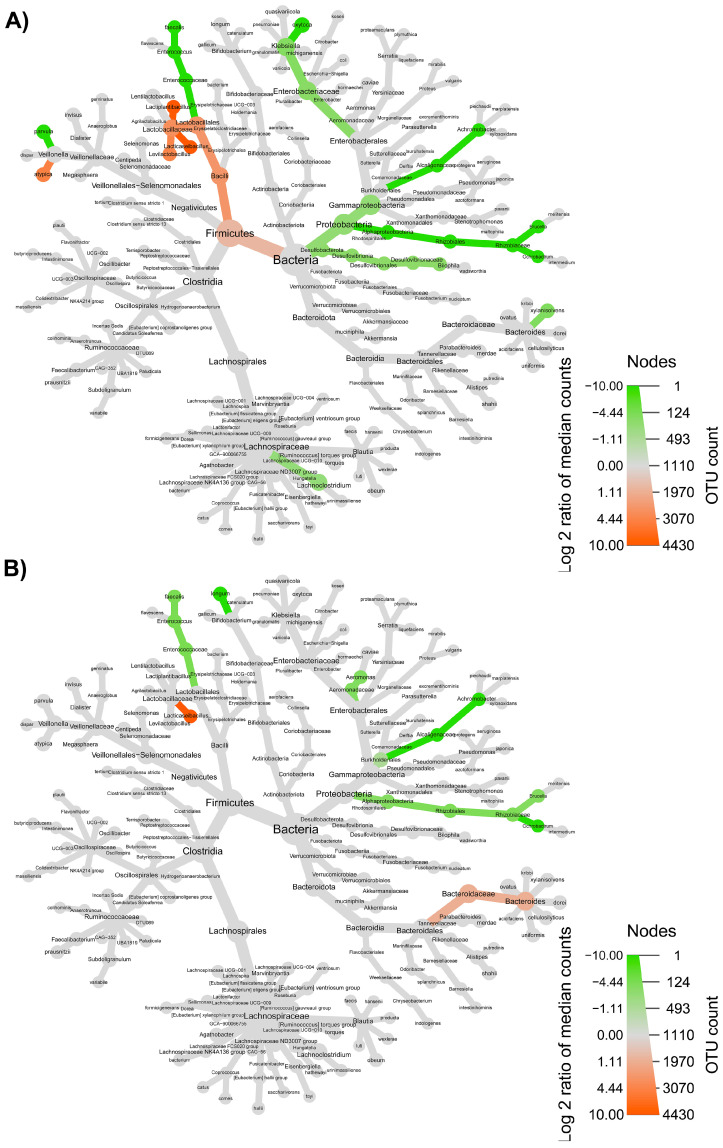
Microbial community taxonomic structure variations at different levels, before compared to after feeding with MRB (**A**), and before feeding against the end of wash out period (**B**). Node size denotes the rarefied ASV count, and colored nodes indicate a significant (*p* < 0.05) fold change for a given taxon: green color means that the taxon is differentially abundant before the treatment, while orange color indicates that the taxon is differentially abundant after feeding (**A**) or after wash out period (**B**). Color intensity is proportional to the fold change.

**Figure 8 nutrients-15-00590-f008:**
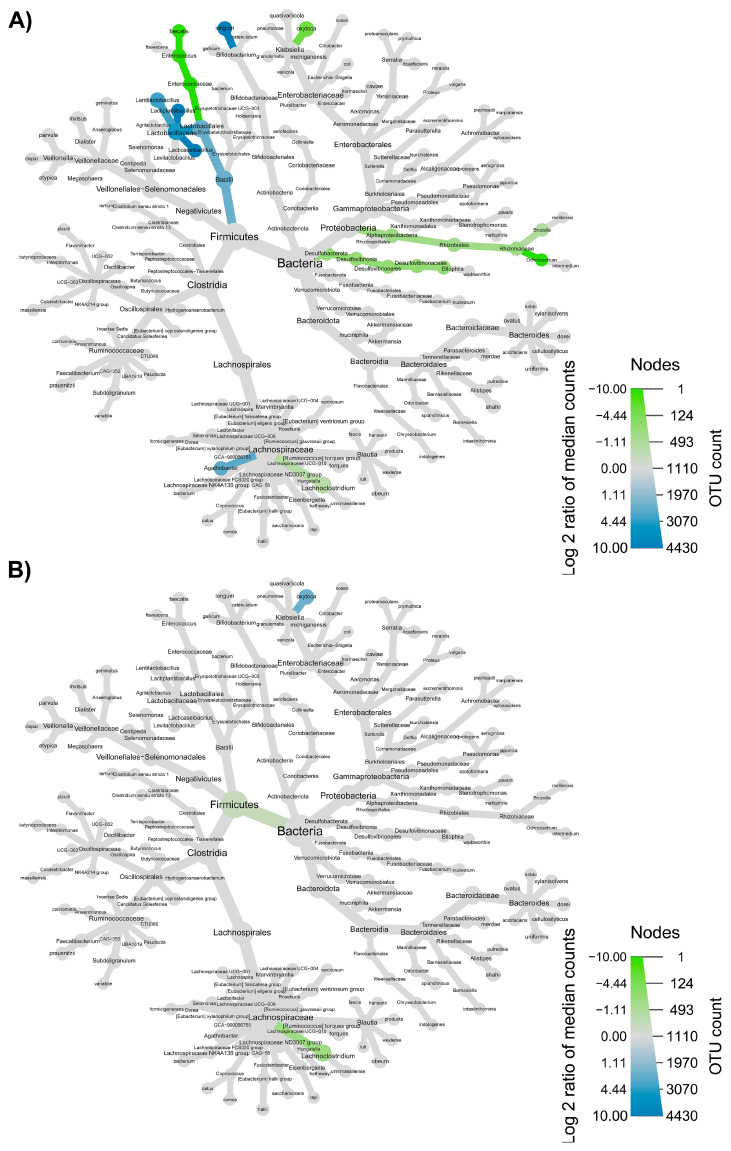
Microbial community taxonomic structure variations at different levels, before compared to after feeding with MWB (**A**), and before feeding against the end of wash out period (**B**). Node size denotes the rarefied ASV count, and colored nodes indicate a significant (*p* < 0.05) fold change for a given taxon: green color means that the taxon is differentially abundant before the treatment, while blue color indicates that the taxon is differentially abundant after feeding (**A**) or after wash out period (**B**). Color intensity is proportional to the fold change.

**Figure 9 nutrients-15-00590-f009:**
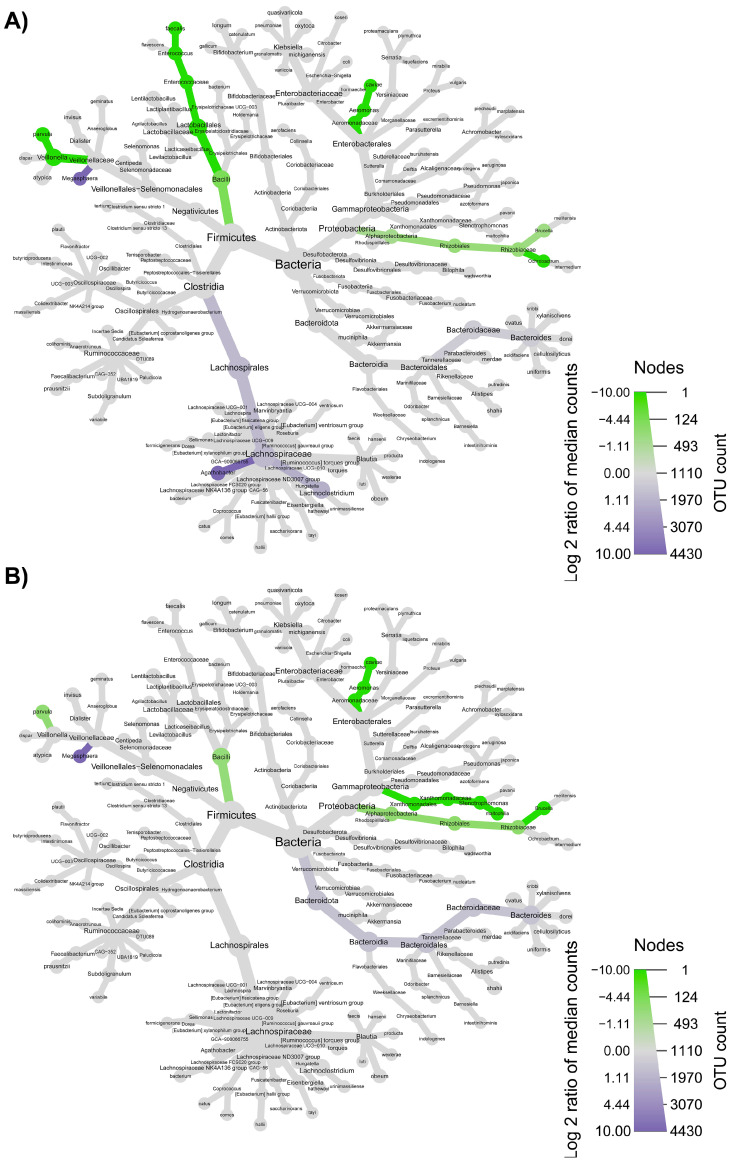
Microbial community taxonomic structure variations at different levels, before compared to after feeding with WB (**A**), and before feeding against the end of wash out period (**B**). Node size denotes the rarefied ASV count, and colored nodes indicate a significant (*p* < 0.05) fold change for a given taxon: green color means that the taxon is differentially abundant before the treatment, while purple color indicates that the taxon is differentially abundant after feeding (**A**) or after wash out period (**B**). Color intensity is proportional to the fold change.

**Figure 10 nutrients-15-00590-f010:**
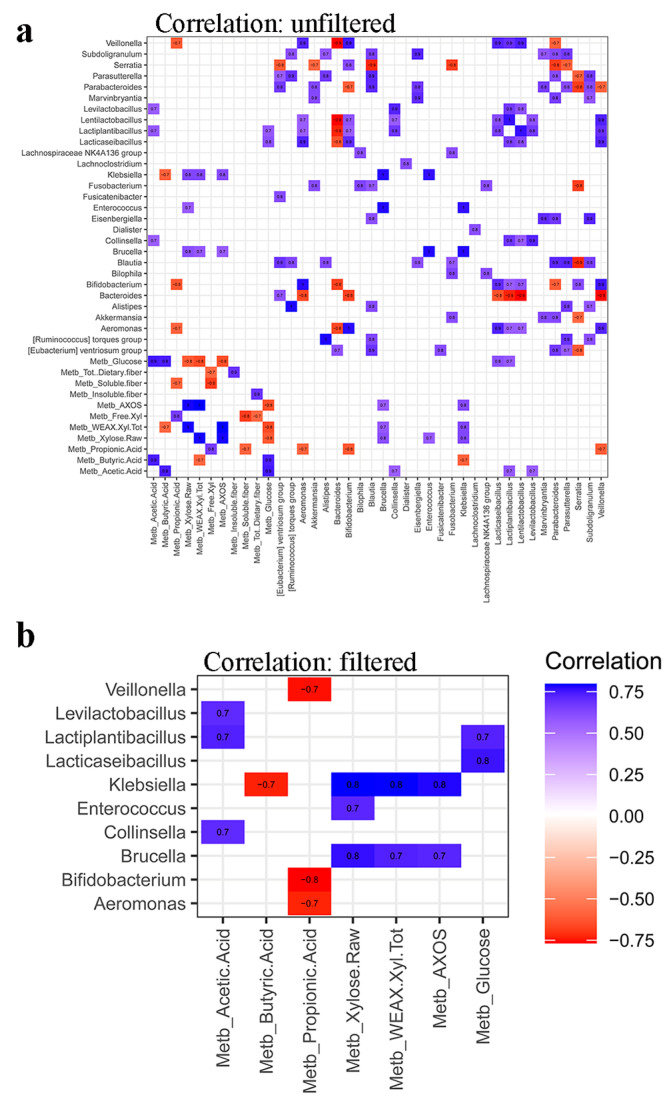
Pearson correlation among initial content of total xylose (Xylose Raw), WEAX Xyl Tot, Free Xyl, AXOS, glucose, short-chain fatty acids (propionic, butyric and acetic acids), insoluble, soluble and total dietary fibers, and microbial abundance in luminal samples after 2 weeks of treatment period. Panel (**a**) shows all and only statistically significant (*p* < 0.05) correlations. Panel (**b**) highlights only significant (*p* < 0.05) correlations between metabolites and microbial groups.

**Table 1 nutrients-15-00590-t001:** Composition of four formulations containing AXOS (MOB, MRB, MWB and WB). Details on the production of AXOS formulations were provided in the Material and Methods section.

	Xyl Tot (g/L)	WEAX Xyl Tot (g/L)	Free Xyl (g/L)	AXOS (g/L)	Insoluble Fibers (g/L)	Soluble Fibers (g/L)	Total Dietary Fibers (g/L)	Glucose (g/L)	Ara/xyl
MOB	82.0 ± 11.3	77.6 ± 9.3	0.6 ± 0.1	72.0 ± 8.5	4.6 ± 0.3	1.2 ± 0.1	5.8 ± 0.2	1.5 ± 0.1	0.06
MRB	14.7 ± 2.3	9.9 ± 0.1	1.5 ± 0.3	10.1 ± 0.9	4.9 ± 0.2	1.0 ± 0.1	6.0 ± 0.3	11.2 ± 0.4	0.34
MWB	20.6 ± 0.5	18.6 ± 1.9	3.1 ± 0.1	16.9 ± 2.7	3.2 ± 0.2	1.0 ± 0.1	4.2 ± 0.2	3.5 ± 0.2	0.25
WB	16.0 ± 0.0	1.3 ± 0.0	0.0 ± 0.0	1.2 ± 0.0	3.4 ± 0.2	1.9 ± 0.1	5.3 ± 0.3	9.6 ± 0.3	0.48

**Table 2 nutrients-15-00590-t002:** SCFAs concentration (mM) in PC and DC reactors before (T0), after one (T1) and two (T2) weeks of treatment period and after two weeks of wash out period (T3). The 4-SHIME^®^ units (1, 2, 3 and 4 units) were treated with MOB, MRB, MWB and WB, respectively.

SCFA (mM)		PC			
		MOB	MRB	MWB	WB
Acetic acid	T0 T1 T2 T3	32.3 ± 1.1 ^a/***c***^ 41.9 ± 11.2 ^c/***b***^ 38.5 ± 10.9 ^d/***b***^ 53.1 ± 2.7 ^a/***a***^	33.8 ± 0.9 ^a/***c***^ 135.7 ± 14.0 ^a/***a***^ 127.1 ± 12.9 ^a/***a***^ 48.3 ± 2.0 ^b/***b***^	32.4 ± 1.0 ^a/***d***^ 83.4 ± 0.3 ^b/***b***^ 90.2 ± 9.3 ^b/***a***^ 42.3 ± 1.9 ^c/***c***^	32.5 ± 1.3 ^a/***c***^ 79.6 ± 12.0 ^b/***a***^ 63.5 ± 13.6 ^c/***b***^ 36.9 ± 5.2 ^d/***c***^
Butyric acid	T0 T1 T2 T3	2.1 ± 0.3 ^a/***b***^ 3.3 ± 0.3 ^c/***a***^ 3.5 ± 0.2 ^d/***a***^ 4.0 ± 0.1 ^a/***a***^	2.5 ± 0.2 ^a/***d***^ 6.8 ± 0.1 ^a/***b***^ 7.7 ± 0.3 ^a/***a***^ 3.2 ± 0.0 ^c/***c***^	2.4 ± 0.7 ^a/***d***^ 5.4 ± 0.0 ^b/***b***^ 7.0 ± 0.3 ^b/***a***^ 3.5 ± 0.1 ^b/***c***^	2.3 ± 0.3 ^a/***c***^ 4.0 ± 0.4 ^c/***a***^ 4.6 ± 1.0 ^c/***a***^ 3.1 ± 0.0 ^c/***b***^
Propionic acid	T0 T1 T2 T3	9.2 ± 1.0 ^a/***d***^ 13.9 ± 0.1 ^c/***c***^ 15.1 ± 0.1 ^c/***a***^ 14.2 ± 0.2 ^a/***b***^	9.1 ± 0.9 ^a/***d***^ 13.1 ± 0.7 ^c/***b***^ 16.3 ± 3.1 ^b/***a***^ 11.7 ± 0.4 ^c/***c***^	9.1 ± 1.2 ^a/***c***^ 15.9 ± 1.3 ^b/***a***^ 12.9 ± 0.6 ^d/***b***^ 12.9 ± 0.5 ^b/***b***^	9.4 ± 0.5 ^a/***d***^ 20.3 ± 1.0 ^a/***a***^ 17.5 ± 0.1 ^a/***b***^ 14.4 ± 0.3 ^a/***c***^
		DC			
		MOB	MRB	MWB	WB
Acetic acid	T0 T1 T2 T3	22.1 ± 0.9 ^a/***d***^ 76.4 ± 4.3 ^b/***a***^ 59.4 ± 10.6 ^c/***b***^ 45.5 ± 1.8 ^b/***c***^	21.7 ± 0.1 ^a/***d***^ 118.1 ± 3.2 ^a/***b***^ 127.6 ± 2.8 ^a/***a***^ 48.3 ± 4.0 ^b/***c***^	22.0 ± 1.1 ^a/***d***^ 76.2 ± 4.8 ^b/***b***^ 92.0 ± 5.9 ^b/***a***^ 52.3 ± 2.5 ^a/***c***^	22.4 ± 1.3 ^a/***d***^ 82.0 ± 5.3 ^b/***a***^ 57.4 ± 4.3 ^c/***b***^ 45.2 ± 4.0 ^b/***c***^
Butyric acid	T0 T1 T2 T3	2.0 ± 0.1 ^a/***c***^ 5.4 ± 0.6 ^b/***a***^ 4.8 ± 0.4 ^c/***a***^ 3.6 ± 0.3 ^b/***b***^	2.0 ± 0.1 ^a/***d***^ 6.1 ± 0.4 ^a/***b***^ 9.9 ± 0.4 ^a/***a***^ 3.3 ± 0.1 ^b/***c***^	2.0 ± 0.2 ^a/***d***^ 5.7 ± 1.1 ^b/***b***^ 8.1 ± 0.7 ^b/***a***^ 4.4 ± 0.1 ^a/***c***^	2.1 ± 0.2 ^a/***c***^ 5.1 ± 0.4 ^c/***a***^ 5.1 ± 0.2 ^c/***a***^ 3.9 ± 0.1 ^b/***b***^
Propionic acid	T0 T1 T2 T3	8.5 ± 1.1 ^a/***c***^ 16.2 ± 0.4 ^b/***a***^ 16.1 ± 0.5 ^c/***a***^ 14.8 ± 0.5 ^a/***b***^	8.8 ± 0.9 ^a/***d***^ 15.3 ± 1.2 ^b/***b***^ 17.3 ± 0.4 ^b/***a***^ 12.8 ± 0.2 ^c/***c***^	8.2 ± 1.5 ^a/***c***^ 16.1 ± 1.3 ^b/***a***^ 15.9 ± 0.8 ^c/***a***^ 13.9 ± 0.4 ^b/***b***^	8.7 ± 1.1 ^a/***d***^ 22.9 ± 0.5 ^a/***a***^ 18.9 ± 0.16 ^a/***b***^ 15.8 ± 0.2 ^a/***c***^

One-way ANOVA and individual post hoc comparisons with the Tukey–Kramer were performed separately for each sampling point and each treatment. For every colon tract (PC and DC), values in the same row with different regular superscript letters differ significantly (*p* < 0.05) (comparison among bioreactors of different 4-SHIME^®^ units at the same sampling point), while values in the same column with different bold/italic superscript letters differ significantly (*p* < 0.05) (comparison among the same bioreactor at different sampling points). The data are the means of three independent analysis ± standard deviations (*n* = 3).

## Data Availability

All sequencing results are publicly available at EML/NCBI/DDBJ under the accession number PRJNA868175 (https://www.ncbi.nlm.nih.gov/bioproject/868175, registration date 10 August 2022).

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
