# Peer review of "The Effect of Hydrolyzed and Fermented Arabinoxylan-Oligo Saccharides (AXOS) Intake on the Middle-Term Gut Microbiome Modulation and Its Metabolic Answer"

_nutrients, 2023, doi:10.3390/nu15030590_

Round 1

Reviewer 1 Report

The manuscript systematically summarized the middle-term dynamics from the intake of hydrolyzed and fermented formulations containing AXOS obtained from three cereal flours. It suggests that hydrolyzed AXOS fermented by lactic acid bacteria could have a more powerful prebiotic effect compared to no hydrolyzed and no fermented wheat bran, shaping the colon microbiome and its metabolism. The manuscript is well structured and the contents are meaningful. However, I have several concerns that need to be addressed to make this outcome more convincing.

1.     Line 12, the background of this study can be briefly summarized in one sentence at the beginning of the abstract.

2.     Line 12, use the full words for the first appearance, followed by the abbreviation. Please check the whole body text.

3.     Line 48, as you said, natural prebiotics were linked with a plethora of beneficial effects leading to significant improvement of health status. So why do you focus on dietary fibers to perform your further exploration?

4.     Line 79, why do you explore the effects of pre-treatments on the prebiotic features of AXOS from oat, rye and wheat brans? Why choose the oat, rye, and wheat brans to get the hydrolyzed AXOS formulation? Please explain. 

5.     Line 85-90, long sentence, please divide into 2 sentences.

6.     Line 105, “WB”? or “wheat bran”? WB means “no fermented wheat bran”, please check carefully.

7.     Line 120, “500 and 800 mL”.

8.     Line 132, “12.5 g/L NaHCO3, 0.9 g/L pancreatin and 6 g/L oxgall”.

9.     Line 135-140, long sentence, please modify.

10.   Line 145, “once a day”.

11.   Line 147, “40 mL”, please check throughout the text.

12.   Line 166, “4 °C” and line 168, “-80 °C”, please check throughout the text.

13.  Line 673, “MOB, MRB, MWB and WB”, use the abbreviation when it appears a second time. Please check and correct the full figure legend.

Author Response

Reviewer 1

The manuscript systematically summarized the middle-term dynamics from the intake of hydrolyzed and fermented formulations containing AXOS obtained from three cereal flours. It suggests that hydrolyzed AXOS fermented by lactic acid bacteria could have a more powerful prebiotic effect compared to no hydrolyzed and no fermented wheat bran, shaping the colon microbiome and its metabolism. The manuscript is well structured and the contents are meaningful.

We thank the Reviewer 1 for the positive comments.

However, I have several concerns that need to be addressed to make this outcome more convincing.

Line 12, the background of this study can be briefly summarized in one sentence at the beginning of the abstract.

Ok, in the newly submitted manuscript, the sentence “Although fermentation and hydrolyzation are well known processes to improve the bioavailability of nutrients and enable the fortification with dietary fibers, the effect of such pre-treatments on the prebiotic features of arabinoxylan-oligosaccharides (AXOS) was not explored yet.” was added (see lines 13-15) to summarize the study background.

Line 12, use the full words for the first appearance, followed by the abbreviation. Please check the whole body text.

Ok, in the newly submitted manuscript, the acronym was defined. See the sentence “…the Simulator of the Human Intestinal Microbial Ecosystem (SHIME)…” at line 16.

Line 48, as you said, natural prebiotics were linked with a plethora of beneficial effects leading to significant improvement of health status. So why do you focus on dietary fibers to perform your further exploration?

Prebiotics are mostly fibers. They are non-digestible food ingredients and beneficially affect the host’s health by selectively stimulating the growth and/or activity of some genera of microorganisms in the colon.

To clarify, in the newly submitted manuscript the sentence was changed as follow “In last decade, natural prebiotics, mostly fibers, earned a considerable importance for human diet as they were linked with a plethora of beneficial effects leading to significant improvement of health status (Benítez-Páez et al., 2019). Even if the exact mechanism of action is not fully known yet, they are fermented by the gut bacteria changing the microbial composition and activity. The abundance of groups/species with capacity for using the variety of simple and complex possible structures of such carbohydrates as energy source increase with resulting beneficial physiologic effect on the host (Müller et al., 2020). The expression of such fermentative machinery in distinct species results in divergent specialization of gut microbiota (Cantu-Jungles and Hamaker, 2020)”. (lines 55-63).

Line 79, why do you explore the effects of pre-treatments on the prebiotic features of AXOS from oat, rye and wheat brans? Why choose the oat, rye, and wheat brans to get the hydrolyzed AXOS formulation? Please explain. 

The effect of fermentation and hydrolyzation pre-treatment was investigated as a large literature demonstrated that such processes are able to increase the bioavailability of compounds/nutrients that are present in the original matrix. For instance, oat, rye, and wheat cereals contain a low amount of AXOS natively and we used the pre-treatment with the xylanase to in-situ increase the amount of AXOS by hydrolyzing the arabinoxylan polymers. As a consequence, this can make the treated matrices more suitable/functional to be used with the purpose to fortify foods and other applications.

Oat, rye, and wheat brans were used for the study as they are among the most frequently used cereals, for instance for bread making.

To clarify, in the newly submitted manuscript the sentence was changed with “Moreover, although fermentation and hydrolyzation are well known processes to improve the bioavailability of AXOS and other nutrients, and to enable the food fortification with dietary fibers (Arora et al., 2021), the effect of such pre-treatments on the prebiotic features of AXOS from the most widely consumed main cereals, like oat, rye and wheat brans, was not explored yet.” (see lines 95-100).

Line 85-90, long sentence, please divide into 2 sentences.

Ok, in the newly submitted manuscript, the long sentence was divided in two shorter sentences (see lines 105-111).

Line 105, “WB”? or “wheat bran”? WB means “no fermented wheat bran”, please check carefully.

Yes, in the newly submitted manuscript, “WB” was replaced with “wheat bran” (see line 126).

Line 120, “500 and 800 mL”.

Ok, in the newly submitted manuscript, “ml” was replaced with “mL”(see line 149).

Line 132, “12.5 g/L NaHCO3, 0.9 g/L pancreatin and 6 g/L oxgall”.

The change was done in the newly submitted manuscript (see lines 162--163).

Line 135-140, long sentence, please modify.

Ok, in the newly submitted manuscript, the long sentence was divided in two shorter sentences (see lines 165-171).

Line 145, “once a day”.

Ok, the change was done in the newly submitted manuscript (see line 176).

Line 147, “40 mL”, please check throughout the text.

Ok, the change was done in the newly submitted manuscript (see line 178), and we checked through the manuscript.

Line 166, “4 °C” and line 168, “-80 °C”, please check throughout the text.

Ok, the change was done in the newly submitted manuscript (see lines 201 and 203), and we checked through the manuscript.

Line 673, “MOB, MRB, MWB and WB”, use the abbreviation when it appears a second time. Please check and correct the full figure legend.

Ok, all figure and table legend were revised accordingly, for all acronyms.

Reviewer 2 Report

Polo et al., did an interesting study on the gut microbiome modulation of the intake of different food that containing AXOS. The manuscript fits the scope of journal of nutrients and is well written. Good job! I only have some minor comments.

1.      Abstract, spell out the SHIME model first.

2.      Line 41, it’s a long list, could the authors cite more papers at the end of the sentence. Similarly, Line 46.

3.      Line 95, could the authors specify what is middle-term dynamics in its first appearance. 

Author Response

Polo et al., did an interesting study on the gut microbiome modulation of the intake of different food that containing AXOS. The manuscript fits the scope of journal of nutrients and is well written. Good job!

We thank the Reviewer 2 for the positive comments.

I only have some minor comments.

Abstract, spell out the SHIME model first.

Ok, in the newly submitted manuscript, the acronym was defined. See the sentence “…the Simulator of the Human Intestinal Microbial Ecosystem (SHIME)…” at line 16.

Line 41, it’s a long list, could the authors cite more papers at the end of the sentence. Similarly, Line 46.

Ok, in the newly submitted manuscript, additional papers were cited at the end of these sentences (see lines 48 and 54).

Line 95, could the authors specify what is middle-term dynamics in its first appearance.

Ok, in the newly submitted manuscript, “over a 2 weeks timespan” was added (see lines 112-113).

Reviewer 3 Report

The issue is very interesting regarding to gut microbiome modulation and metabolic effects of human intaking hydrolyzed and fermented arabi-3 noxylan-oligosaccharides (AXOS).

l   I suggested that title can be changed. For example, “The effect of intaking AXOS on middle-term gut microbiome ……”.

 Figure 1.4 In the experimental design, what is the purpose of wash out period?

l   Line 558-In general, during the wash out period the gut microbiome modulated again to a final 558 (after 2 weeks from intake interruption) structure that is more similar to the original con dition, even if some variations still persist. Suggested the “structure” in the sentence should be changed.

l   Line475-Synergy among AXOS composition, microbiome, SCFA and dietary fibers. I suggested a figure or table shows correlation among AXOS composition, microbiome, SCFA and dietary fibers.

Author Response

The issue is very interesting regarding to gut microbiome modulation and metabolic effects of human intaking hydrolyzed and fermented arabi-3 noxylan oligosaccharides (AXOS).

We thank the Reviewer 3 for the positive comments.

I suggested that title can be changed. For example, “The effect of intaking AXOS on middle-term gut microbiome ……”.

Ok, in the newly submitted manuscript, the title was changed with “The effect of hydrolyzed and fermented arabinoxylan-oligosaccharides (AXOS) intake on the middle-term gut microbiome modulation and its metabolic answer” (see lines 2-4).

 Figure 1. In the experimental design, what is the purpose of wash out period?

A wash out period was included in the experimental design to investigate if the changes in colon ecosystems (that are induced by the intake of MOB, MRB, MWB and WB) persist after the intake interruption.

Line 558-In general, during the wash out period the gut microbiome modulated again to a final 558 (after 2 weeks from intake interruption) structure that is more similar to the original condition, even if some variations still persist. Suggested the “structure” in the sentence should be changed.

Ok, in the newly submitted manuscript, the term “structure” was replaced with “composition” (see line 581).

Line475-Synergy among AXOS composition, microbiome, SCFA and dietary fibers. I suggested a figure or table shows correlation among AXOS composition, microbiome, SCFA and dietary fibers.

The correlation is showed by the figure 10.

Round 2

Reviewer 3 Report

N/A